# Scalable Coordinated Exploration in Concurrent Reinforcement Learning

**Maria Dimakopoulou**
Stanford University
madima@stanford.edu

**Ian Osband**
Google DeepMind
iosband@google.com

**Benjamin Van Roy**
Stanford University
bvr@stanford.edu

## Abstract

We consider a team of reinforcement learning agents that concurrently operate in a common environment, and we develop an approach to efficient coordinated exploration that is suitable for problems of practical scale. Our approach builds on seed sampling[1] and randomized value function learning [11]. We demonstrate that, for simple tabular contexts, the approach is competitive with previously proposed tabular model learning methods [1]. With a higher-dimensional problem and a neural network value function representation, the approach learns quickly with far fewer agents than alternative exploration schemes.

## 1 Introduction

Consider a farm of robots operating concurrently, learning how to carry out a task, as studied in [3]. There are benefits to scale, since a larger number of robots can gather and share larger volumes of data that enable each to learn faster. These benefits are most dramatic if the robots explore in a coordinated fashion, diversifying their learning goals and adapting appropriately as data is gathered. Web services present a similar situation, as considered in [18]. Each user is served by an agent, and the collective of agents can accelerate learning by intelligently coordinating how they experiment. Considering its importance, the problem of coordinated exploration in reinforcement learning has received surprisingly little attention; while [3] and [18] consider teams of agents that gather data in parallel, they do not address coordination of data gathering, though this can be key to team performance. Dimakopolou and Van Roy [1] recently identified properties that are essential to efficient coordinated exploration and proposed suitable tabular model learning methods based on *seed sampling*. Though this represents a conceptual advance, the methods do not scale to meet the needs of practical applications, which require generalization to address intractable state spaces. In this paper, we develop scalable reinforcement learning algorithms that aim to efficiently coordinate exploration and we present computational results that establish their substantial benefit.

Work on coordinated exploration builds on a large literature that addresses efficient exploration in single-agent reinforcement learning (see, e.g., [6, 5, 21]). A growing segment of this literature studies and extends posterior sampling for reinforcement learning (PSRL) [19], which has led to statistically efficient and computationally tractable approaches to exploration [10, 12, 13]. The methods we will propose leverage this line of work, particularly the use of randomized value function learning [14].

The problem we address is known as concurrent reinforcement learning [18, 15, 4, 16, 1]. A team of reinforcement learning agents interact with the same unknown environment, share data with one another, and learn in parallel how to operate effectively. To learn efficiently in such settings, the agents should coordinate their exploratory effort. Three properties essential to efficient coordinated exploration, identified in [1], are real-time *adaptivity* to shared observations, *commitment* to carry through with action sequences that reveal new information, and *diversity* across learning opportunities pursued by different agents. That paper demonstrated that upper-confidence-bound (UCB) exploration schemes for concurrent reinforcement learning (concurrent UCRL), such as those

discussed in [15, 4, 16], fail to satisfy the diversity property due to their deterministic nature. Further, a straightforward extension of PSRL to the concurrent multi-agent setting, in which each agent independently samples a new MDP at the start of each time period, as done in [7], fails to satisfy the commitment property because the agents are unable to explore the environment thoroughly [17]. As an alternative, [1] proposed seed sampling, which extends PSRL in a manner that simultaneously satisfies the three properties. The idea is that each concurrent agent independently samples a random seed, a mapping from seed to the MDP is determined by the prevailing posterior distribution. Independence among seeds diversifies exploratory effort among agents. If the mapping is defined in an appropriate manner, the fact that each agent maintains a consistent seed ensures a sufficient degree of commitment, while the fact that the posterior adapts to new data allows each agent to react intelligently to new information.

Algorithms presented in [1] are tabular and hence do not scale to address intractable state spaces. Further, computational studies carried out in [1] focus on simple stylized problems designed to illustrate the benefits of seed sampling. In the next section, we demonstrate that observations made in these stylized contexts extend to a more realistic problem involving swinging up and balancing a pole. Subsequent sections extend the seed sampling concept to operate with generalizing randomized value functions [14], leading to new algorithms such as seed temporal-difference learning (*seed TD*) and seed least-squares value iteration (*seed LSVI*). We show that on tabular problems, these scalable seed sampling algorithms perform as well as the tabular seed sampling algorithms of [1]. Finally, we present computational results demonstrating effectiveness of one of our new algorithms applied in conjunction with a neural network representation of the value function on another pole balancing problem with a state space too large to be addressed by tabular methods.

## 2 Seeding with Tabular Representations

This section shows that the advantages of seed sampling over alternative exploration schemes extend beyond the toy problems with known transition dynamics and a handful of unknown rewards considered in [1]. We consider a problem that is more realistic and complex, but of sufficiently small scale to be addressed by tabular methods, in which a group of agents learn to swing-up and balance a pole. We demonstrate that seed sampling learns to achieve the goal quickly and with far fewer agents than other exploration strategies.

In the classic problem [20], a pole is attached to a cart that moves on a frictionless rail. We modify the problem so that deep exploration is crucial to identifying rewarding states and thus learning the optimal policy. Unlike the traditional cartpole problem, where the interaction begins with the pole stood upright and the agent must learn to balance it, in our problem the interaction begins with the pole hanging down and the agent must learn to swing it up. The cart moves on an infinite rail. Concretely the agent interacts with the environment through the state $s_t = (\phi_t, \dot{\phi}_t) \in \Re^2$, where $\phi_t$ is the angle of the pole from the vertical, upright position $\phi = 0$ and $\dot{\phi}_t$ is the respective angular velocity. The cart is of mass $M = 1$ and the pole has mass $m = 0.1$ and length $l = 1$, with acceleration due to gravity $g = 9.8$. At each timestep the agent can apply a horizontal force $F_t$ to the cart. The second order differential equation governing the system is $\ddot{\phi}_t = \frac{g \sin(\phi_t) - \cos(\phi_t)\tau_t}{\frac{l}{2}\left(\frac{4}{3} - \frac{m}{m+M}\cos(\phi_t)^2\right)}$, $\tau_t = \frac{F_t + \frac{l}{2}\dot{\phi}_t^2 \sin(\phi_t)}{m+M}$
[11]. We discretize the evolution of this second order differential equation with timescale $\Delta t = 0.02$ and present a choice of actions $F_t = \{-10, 0, 10\}$ for all $t$. At each timestep the agent pays a cost $\frac{|F_t|}{1000}$ for its action but can receive a reward of 1 if the pole is balanced upright ($\cos(\phi_t) > 0.75$) and steady (angular velocity less than 1). The interaction ends after 1000 actions, i.e. at $t = 20$. The environment is modeled as a time-homogeneous MDP, which is identified by $\mathcal{M} = (\mathcal{S}, \mathcal{A}, \mathcal{R}, \mathcal{P}, \rho)$, where $\mathcal{S}$ is the discretized state space $[0, 2\pi] \times [-2\pi, 2\pi]$, $\mathcal{A} = \{-10, 0, 10\}$ is the action space, $\mathcal{R}$ is the reward model, $\mathcal{P}$ is the transition model and $\rho$ is the initial state distribution.

Consider a group of $K$ agents, who explore and learn to operate in parallel in this common environment. Each $k$th agent begins at state $s_{k,0} = (\pi, 0) + w_k$, where each component of $w_k$ is uniformly distributed in $[-0.05, 0.05]$. Each agent $k$ takes an action at arrival times $t_{k,1}, t_{k,2}, \dots, t_{k,H}$ of an independent Poisson process with rate $\kappa = 1$. At time $t_{k,m}$, the agent takes action $a_{k,m}$, transitions from state $s_{k,m-1}$ to state $s_{k,m}$ and observes reward $r_{k,m}$. The agents are uncertain about the transition structure $\mathcal{P}$ and share a common Dirichlet prior over the transition probabilities associated with each state-action pair $(s, a) \in \mathcal{S} \times \mathcal{A}$ with parameters $\alpha_0(s, a, s') = 1$, for all $s' \in \mathcal{S}$. The agents are also uncertain about the reward structure $\mathcal{R}$ and share a common Gaussian prior over the reward

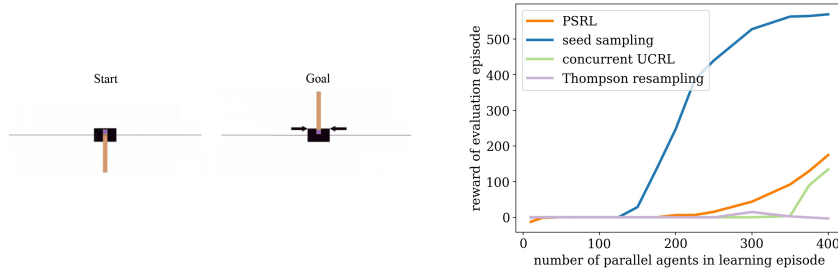

Figure 1: Performance of PSRL (no adaptivity), concurrent UCRL (no diversity), Thompson resampling (no commitment) and seed sampling in the tabular problem of learning how to swing and keep upright a pole attached to a cart that moves left and right on an infinite rail.

associated with each state-action pair $(s, a) \in \mathcal{S} \times \mathcal{A}$ with parameters $\mu_0(s, a) = 0, \sigma_0^2(s, a) = 1$. Agents share information in real time and update their posterior beliefs.

We compare seed sampling with three baselines, PSRL, concurrent UCRL and Thompson resampling. In PSRL, each agent $k$ samples an MDP $\mathcal{M}_{k,0}$ from the common prior at time $t_{k,0}$ and computes the optimal policy $\pi_{k,0}(\cdot)$ with respect to $\mathcal{M}_{k,0}$, which does not change throughout the agent's interaction with the environment. Therefore, the PSRL agents *do not adapt* to the new information in real-time. On the other hand, in concurrent UCRL, Thompson resampling and seed sampling, at each time $t_{k,m}$, the agent $k$ generates a new MDP $\mathcal{M}_{k,m}$ based on the data gathered by all agents up to that time, computes the optimal policy $\pi_{k,m}$ for $\mathcal{M}_{k,m}$ and takes an action $a_{k,m} = \pi_{k,m}(s_{k,m-1})$ according to the new policy. Concurrent UCRL is a deterministic approach according to which all the parallel agents construct the same optimistic MDP conditioned on the common shared information up to that time. Therefore, the concurrent UCRL agents *do not diversify* their exploratory effort. Thompson resampling has each agent independently sample a new MDP at each time period from the common posterior distribution conditioned on the shared information up to that time. Resampling an MDP independently at each time period breaks the agent's intent to pursue a sequence of actions revealing the rare reward states. Therefore, the Thompson resampling agents *do not commit*. Finally, in seed sampling, at the beginning of the experiment, each agent $k$ samples a random seed $\omega_k$ with two components that remain fixed throughout the experiment. The first component is $|\mathcal{S}|^2|\mathcal{A}|$ sequences of independent and identically distributed $\mathrm{Exp}(1)$ random variables; the second component is $|\mathcal{S}||\mathcal{A}|$ independent and identically distributed $\mathcal{N}(0, 1)$ random variables. At each time $t_{k,m}$, agent $k$ maps the data gathered by all agents up to that time and its seed $\omega_k$ to an MDP $\mathcal{M}_{k,m}$ by combining the Exponential-Dirichlet seed sampling and the standard-Gaussian seed sampling methods described in [1]. Independence among seeds diversifies exploratory effort among agents. The fact that the agent maintains a consistent seed leads to a sufficient degree of commitment, while the fact that the posterior adapts to new data allows the agent to react intelligently to new information.

After the end of the learning interaction, there is an evaluation of what the group of $K$ agents learned. The performance of each algorithm is measured with respect to the reward achieved during this evaluation, where a greedy agent starts at $s_0 = (\pi, 0)$, generates the expected MDP of the cartpole environment based on the posterior beliefs formed by the $K$ parallel agents at the end of their learning, and interacts with the cartpole as dictated by the optimal policy with respect to this MDP. Figure 1 plots the reward achieved by the evaluation agent for increasing number of PSRL, seed sampling, concurrent UCRL and Thompson resampling agents operating in parallel in the cartpole environment. As the number of parallel learning agents grows, seed sampling quickly increases its evaluation reward and soon attains a high reward only within 20 seconds of learning. On the other hand, the evaluation reward achieved by episodic PSRL (no adaptivity), concurrent UCRL (no diversity), and Thompson resampling (no commitment) does not improve at all or improves in a much slower rate as the number of parallel agents increases.

## 3   Seeding with Generalizing Representations

As we demonstrated in Section 2, seed sampling can offer great advantage over other exploration schemes. However, our examples involved tabular learning and the algorithms we considered do

not scale gracefully to address practical problems that typically pose enormous state spaces. In this section, we propose an algorithmic framework that extends the seeding concept from tabular to generalizing representations. This framework supports scalable reinforcement learning algorithms with the degrees of adaptivity, commitment, and intent required for efficient coordinated exploration.

We consider algorithms with which each agent is instantiated with a seed and then learns a parameterized value function over the course of operation. When data is insufficient, the seeds govern behavior. As data accumulates and is shared across agents, each agent perturbs each observation in a manner distinguished by its seed before training its value function on the data. The varied perturbations of shared observations result in diverse value function estimates and, consequently, diverse behavior. By maintaining a constant seed throughout learning, an agent does not change his interpretation of the same observation from one time period to the next, and this achieves the desired level of commitment, which can be essential in the presence of delayed consequences. Finally, by using parameterized value functions, agents can cope with intractably large state spaces. Section 3.1 offers a more detailed description of our proposed algorithmic framework, and Section 3.2 provides examples of algorithms that fit this framework.

## 3.1 Algorithmic Framework

There are $K$ agents, indexed $1, \ldots, K$. The agents operate over $H$ time periods in identical environments, each with state space $\mathcal{S}$ and action space $\mathcal{A}$. Denote by $t_{k,m}$ the time at which agent $k$ applies its $m$th action. The agents may progress synchronously ($t_{k,m} = t_{k',m}$) or asynchronously ($t_{k,m} \neq t_{k',m}$). Each agent $k$ begins at state $s_{k,0}$. At time $t_{k,m}$, agent $k$ is at state $s_{k,m}$, takes action $a_{k,m}$, observes reward $r_{k,m}$ and transitions to state $s_{k,m+1}$. In order for the agents to adapt their policies in real-time, each agent has access to a buffer $\mathcal{B}$ with observations of the form $(s, a, r, s')$. This buffer stores past observations of all $K$ agents. Denote by $\mathcal{B}_t$ the content of this buffer at time $t$. With value function learning, agent $k$ uses a family $\tilde{Q}_k$ of state action value functions indexed by a set of parameters $\Theta_k$. Each $\theta \in \Theta_k$ defines a state-action value function $\tilde{Q}_{k,\theta} : \mathcal{S} \times \mathcal{A} \to \Re$. The value $\tilde{Q}_{k,\theta}(s, a)$ could be, for example, the output of a neural network with weights $\theta$ in response to an input $(s, a)$. Initially, the agents may have prior beliefs over the parameter $\theta$, such as the expectation, $\bar{\theta}$, or the level of uncertainty, $\lambda$, on $\theta$.

Agents diversify their behavior through a seeding mechanism. Under this mechanism, each agent $k$ is instantiated with a seed $\omega_k$. Seed $\omega_k$ is intrinsic to agent $k$ and differentiates how agent $k$ interprets the common history of observations in the buffer $\mathcal{B}$. A form of seeding is that each agent $k$ can independently and randomly perturb observations in the buffer. For example, different agents $k, k'$ can add different noise terms $z_{k,j}$ and $z_{k',j}$ of variance $v$, which are determined by seeds $\omega_k$ and $\omega_{k'}$, respectively, to rewards from the same $j$th observation $(s_j, a_j, r_j, s'_j)$ in the buffer $\mathcal{B}$, as discussed in [14] for the single-agent setting. This induces diversity by creating modified training sets from the same history among the agents. Based on the prior distribution for the parameter $\theta$, agent $k$ can initialize the value function with a sample $\hat{\theta}_k$ from this distribution, with the seed $\omega_k$ providing the source of randomness. These independent value function parameter samples diversify the exploration in initial stages of operation. The seed $\omega_k$ remains fixed throughout the course of learning. This induces a level of commitment in agent $k$, which can be important in reinforcement learning settings where delayed consequences are present.

At time $t_{k,m}$, before taking the $m$th action, agent $k$ fits its generalized representation model on the history (or a subset thereof) of observations $(s_j, a_j, r_j, s'_j)$ perturbed by the noise seeds $z_{k,j}$, $j = 1, \ldots, |\mathcal{B}_{t_{k,m}}|$. The initial parameter seed $\hat{\theta}_k$ can also play a role in subsequent stages of learning, other than the first time period, by influencing the model fitting. An example of employing the initial parameter seed $\hat{\theta}_k$ in the model fitting of subsequent time periods is by having a function $\psi(\cdot)$ as a regularization term in which $\hat{\theta}_k$ appears. By this model fitting, agent $k$ obtains parameters $\theta_{k,m}$ at time period $t_{k,m}$. These parameters define a state-action value function $\tilde{Q}_{k,\theta_{k,m}}(\cdot, \cdot)$ based on which a policy is computed. Based on the obtained policy and its current state $s_{k,m}$, the agent takes a greedy action $a_{k,m}$, observes reward $r_{k,m}$ and transitions to state $s_{k,m+1}$. The agent $k$ stores this observation $(s_{k,m}, a_{k,m}, r_{k,m}, s_{k,m+1})$ in the buffer $\mathcal{B}$ so that all agents can access it next time they fit their models. For learning problems with large learning periods, it may be practical to cap the common buffer to a certain capacity $C$ and once this capacity is exceeded to start overwriting observations at

random. In this case, the way observations are overwritten can also be different for each agent and determined by seed $\omega_k$ (e.g. by $\omega_k$ also defining random permutation of indices $1, \dots, C$).

The ability of the agents to make decisions in the high-dimensional environments of real systems, where the number of states is enormous or even infinite, is achieved through the value function representations, while coordinating the exploratory effort of the group of agents is achieved through the way that the seeding mechanism controls the fitting of these generalized representations. As the number of parallel agents increases, this framework enables the agents to learn to operate and achieve high rewards in complex environments very quickly.

## 3.2 Examples of Algorithms

We now present examples of algorithms that fit the framework of Section 3.1.

In our proposed algorithms, agents share a Gaussian prior over unknown parameters $\theta^* \sim \mathcal{N}(\bar{\theta}, \lambda I)$ and a Gaussian likelihood, $\mathcal{N}(0, v)$. Each agent $k$ samples independently noise seeds $z_{k,j} \sim \mathcal{N}(0, v)$ for each observation $j$ in the buffer and initial parameter seeds $\hat{\theta}_k \sim \mathcal{N}(\bar{\theta}, \lambda I)$. These seeds remain fixed throughout learning. We now explain how the algorithms we propose satisfy the three properties of efficient coordinated exploration.

1. *Adaptivity*: The key idea behind randomized value functions is that fitting a model to a randomly perturbed prior and randomly perturbed observations can be used to generate posterior samples or approximate posterior samples. Consider the data $(X, y) = \left(\{x_j\}_{j=1}^N, \{y_j\}_{j=1}^N\right)$, where $y_j = \theta^{*T} x_j + \epsilon_j$, with IID $\epsilon_j \sim \mathcal{N}(0, v)$. Let $f_\theta = \theta^T x$, $\hat{\theta} \sim \mathcal{N}(\bar{\theta}, \lambda I)$ and $z_j \sim \mathcal{N}(0, v)$. Then, the solution to $\text{argmin}_\theta \left( \frac{1}{v} \sum_j \left( y_j + z_j - f_\theta(x_i) \right)^2 + \frac{1}{\lambda} \|\theta - \hat{\theta}\|_2^2 \right)$ is a sample from the posterior of $\theta^*$ given $(X, y)$ [14]. This sample can be computed for non-linear $f_\theta$ as well, although it will not be from the exact posterior. In the concurrent setting, when each agent $k$ draws initial parameter seed $\hat{\theta}_k \sim \mathcal{N}(\bar{\theta}, \lambda I)$ and noise seeds $z_{k,1}, z_{k,2}, \dots \sim \mathcal{N}(0, v)$ at each time period it can solve this value-function optimization problem to obtain a posterior parameter sample based on the high-dimensional observations gathered by all agents so far.

2. *Diversity*: The independence of the initial parameter seeds $\hat{\theta}_k$ and noise seeds $z_{k,j}$ among agents diversifies exploration both when there are no available observations and when the agents have access to the same shared observations.

3. *Commitment*: Each agent $k$ applies the same perturbation $z_{k,j}$ to each $j$th observation and uses the same regularization $\hat{\theta}_k$ throughout learning; this provides the requisite level of commitment.

### 3.2.1 Seed Least Squares Value Iteration (Seed LSVI)

LSVI computes a sequence of value functions parameters reflecting optimal expected rewards over an expanding horizon based on observed data. In seed LSVI, each $k$th agent's initial parameter $\theta_{k,0}$ is set to $\hat{\theta}_k$. Before its $m$th action, agent $k$ uses the buffer of observations gathered by all $K$ agents up to that time, or a subset thereof, and the random noise terms $z_k$ to carry out LSVI, initialized with $\tilde{\theta}_H = 0$, where $H$ is the LSVI planning horizon:

$$\tilde{\theta}_h = \underset{\theta}{\text{argmin}} \left( \frac{1}{v} \sum_{(s_j, a_j, r_j, s_j')} \left( r_j + \max_{a \in \mathcal{A}} \tilde{Q}_{k, \tilde{\theta}_{h+1}}(s_j', a) + z_{k,j} - \tilde{Q}_{k,\theta}(s_j, a_j) \right)^2 + \psi(\theta, \hat{\theta}_k) \right)$$

for $h = H - 1, \dots, 0$, where $\psi(\theta, \hat{\theta}_k)$ is a regularization penalty (e.g. $\psi(\theta, \hat{\theta}_k) = \frac{1}{\lambda} \|\theta - \hat{\theta}_k\|_2^2$). After setting $\theta_{k,m} = \tilde{\theta}_0$, agent $k$ applies action $a_{k,m} = \text{argmax}_{a \in \mathcal{A}} \tilde{Q}_{k,\theta_{k,m}}(s_{k,m}, a)$. Note that the agent's random seed can be viewed as $\omega_k = (\hat{\theta}_k, z_{k,1}, z_{k,2}, \dots)$.

### 3.2.2 Seed Temporal-Difference Learning (Seed TD)

When the dimension of $\theta$ is very large, significant computational time may be required to produce an estimate with LSVI, and using first-order algorithms in the vein of stochastic gradient descent, such as TD, can be beneficial. In seed TD, each $k$th agent's initial parameter $\theta_{k,0}$ is set to $\hat{\theta}_k$. Before its

$m$th action, agent $k$ uses the buffer of observations gathered by all $K$ agents up to that time to carry out $N$ iterations of stochastic gradient descent, initialized with $\tilde{\theta}_0 = \theta_{k,m-1}$:

$$\tilde{\theta}_n = \tilde{\theta}_{n-1} - \alpha \nabla_\theta \mathcal{L}(\tilde{\theta}_{n-1})$$

$$\mathcal{L}(\theta) = \frac{1}{v} \sum_{(s_j, a_j, r_j, s'_j)} \left( r_j + \gamma \max_{a \in \mathcal{A}} \tilde{Q}_{k, \tilde{\theta}_{n-1}}(s'_j, a) + z_{k,j} - \tilde{Q}_{k,\theta}(s_j, a_j) \right)^2 + \psi(\theta, \hat{\theta}_k)$$

for $n = 1, \ldots, N$, where $\alpha$ is the TD learning rate, $\mathcal{L}(\theta)$ is the loss function, $\gamma$ is the discount rate and $\psi(\theta, \hat{\theta}_k)$ is a regularization penalty (e.g. $\psi(\theta, \hat{\theta}_k) = \frac{1}{\lambda} \|\theta - \hat{\theta}_k\|_2^2$). After setting $\theta_{k,m} = \tilde{\theta}_N$, agent $k$ applies action $a_{k,m} = \mathrm{argmax}_{a \in \mathcal{A}} \tilde{Q}_{k, \theta_{k,m}}(s_{k,m}, a)$. Note that the agent's random seed can be viewed as $\omega_k = (\hat{\theta}_k, z_{k,1}, z_{k,2}, \ldots)$.

### 3.2.3 Seed Ensemble

When the number of parallel agents is large, instead of having each one of the $K$ agents fit a separate value function model (e.g. $K$ separate neural networks), we can have an ensemble of $E$ models, $E < K$, to decrease computational requirements. Each model $e = 1, \ldots, E$ is initialized with $\hat{\theta}_e \sim \mathcal{N}(\bar{\theta}, \lambda)$ from the common prior belief on parameters $\theta$, which is fixed and specific to model $e$ of the ensemble. Moreover model $e$ is trained on the buffer of observations $\mathcal{B}$ according to one of the methods of Section 3.2.1 or 3.2.2. Each observation $(s_j, a_j, r_j, s'_j) \in \mathcal{B}$ is perturbed with noise $z_{e,j}$, which is also fixed and specific to model $e$ of the ensemble. Note that the agent's $k$ random seed, $\omega_k$, is a randomly drawn index $e = 1, \ldots, E$ associated with a model from the ensemble.

### 3.2.4 Extensions

The framework we propose is not necessarily constrained to value function approximation methods. For instance, one could use the same principles for policy function approximation, where each agent $k$ defines a policy function $\tilde{\pi}_k(s, a, \theta)$ and before its $m$th action uses the buffer of observations gathered by all $K$ agents up to that time and its seeds $z_k$ to perform policy gradient.

## 4 Computational Results

In this section, we present computational results that demonstrate the robustness and effectiveness of the approach we suggest in Section 3. In Section 4.1, we present results that serve as a sanity check for our approach. We show that in the tabular toy problems considered in [1], seeding with generalized representations performs equivalently with the seed sampling algorithm proposed in [1], which is particularly designed for tabular settings and can benefit from very informative priors. In Section 4.2, we scale-up to a high-dimensional problem, which would be too difficult to address by any tabular approach. We use the concurrent reinforcement learning algorithm of Sections 3.2.2 and 3.2.3 with a neural network value function approximation and we see that our approach explores quickly and achieves high rewards.

### 4.1 Sanity Checks

The authors of [1] considered two toy problems that demonstrate the advantage of seed sampling over Thompson resampling or concurrent UCRL. We compare the performance of seed LSVI (Section 3.2.1) and seed TD (Section 3.2.2), which are designed for generalized representations, with seed sampling, Thompson resampling and concurrent UCRL which are designed for tabular representations.

The first toy problem is the "bipolar chain" of figure 2a. The chain has an even number of vertices, $N$, $\mathcal{V} = \{0, 1, \ldots, N - 1\}$ and the endpoints are absorbing. From any inner vertex of the chain, there are two edges that lead deterministically to the left or to the right. The leftmost edge $e_L = (1, 0)$ has weight $\theta_L$ and the rightmost edge $e_R = (N - 2, N - 1)$ has weight $\theta_R$, such that $|\theta_L| = |\theta_R| = N$ and $\theta_R = -\theta_L$. All other edges have weight $\theta_e = -0.1$. Each one of the $K$ agents starts from vertex $N/2$, and its goal is to maximize the accrued reward. We let the agents interact with the environment for $2N$ time periods. As in [1], seed sampling, Thompson resampling and concurrent UCRL, know everything about the environment except from whether $\theta_L = N, \theta_R = -N$ or $\theta_L = -N, \theta_R = N$

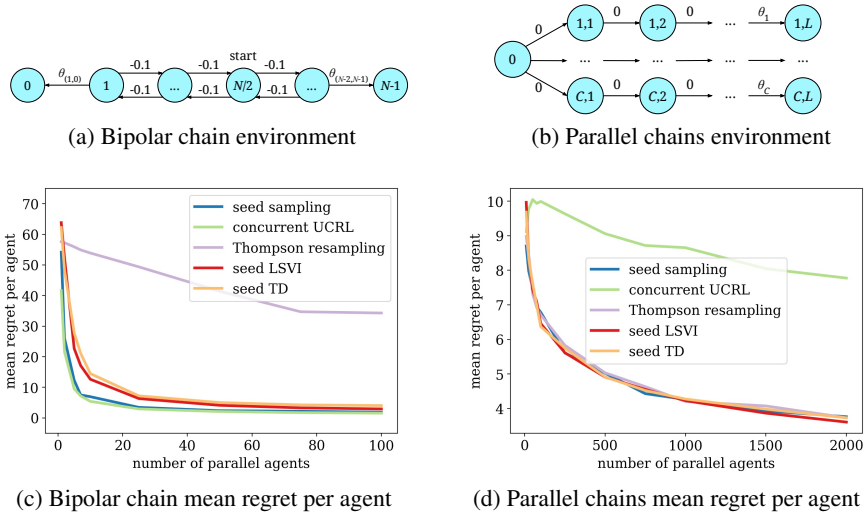

(a) Bipolar chain environment

(b) Parallel chains environment

(c) Bipolar chain mean regret per agent

(d) Parallel chains mean regret per agent

Figure 2: Comparison of the scalable seed algorithms, seed LSVI and seed TD, with their tabular counterpart seed sampling and the tabular alternatives concurrent UCRL and Thompson resampling in the toy settings considered in [1]. This comparison serves as a sanity check.

and they share a common prior that assigns probability $p = 0.5$ to either scenario. Once an agent reaches either of the endpoints, all $K$ agents learn the true value of $\theta_L$ and $\theta_R$. Seed LSVI and seed TD use $N$-dimensional one-hot encoding to represent any of the chain's states and a linear value function representation. Unlike, the tabular algorithms, seed LSVI and seed TD start with a completely uninformative prior. We run the algorithms with different number of parallel agents $K$ operating on a chain with $N = 50$ vertices. Figure 2c shows the mean reward per agent achieved as $K$ increases. The "bipolar chain" example aims to highlight the importance of the commitment property. As explained in [1], concurrent UCRL and seed sampling are expected to perform in par because they exhibit commitment, but Thompson resampling is detrimental to exploration because resampling a MDP in every time period leads the agents to oscillation around the start vertex. Seed LSVI and seed TD exhibit commitment and perform almost as well as seed sampling, which not only is designed for tabular problems but also starts with a significantly more informed prior.

The second toy problem is the "parallel chains" of figure 2b. Starting from vertex 0, each of the $K$ agents chooses one of the $c = 1, \ldots, C$ chains, of length $L$. Once a chain is chosen, the agent cannot switch to another chain. All the edges of each chain $c$ have zero weights, apart from the edge incoming to the last vertex of the chain, which has weight $\theta_c \sim \mathcal{N}(0, \sigma_0^2 + c)$. The objective is to choose the chain with the maximum reward. As in [1], seed sampling, Thompson resampling and concurrent UCRL, know everything about the environment except from $\theta_c, \forall c = 1, \ldots, C$, on which they share a common, well-specified prior. Once an agent traverses the last edge of chain $c$, all agents learn $\theta_c$. Seed LSVI and seed TD use $N$-dimensional one-hot encoding to represent any of the chain's states and a linear value function representation. As before, seed LSVI and seed TD start with a completely uninformative prior. We run the algorithms with different number of parallel agents $K$ operating on a parallel chain environment with $C = 4$, $L = 4$ and $\sigma_0^2 = 100$. Figure 2d shows the mean reward per agent achieved as $K$ increases. The "parallel chains" example aims to highlight the importance of the diversity property. As explained in [1], Thompson resampling and seed sampling are expected to perform in par because they diversify, but concurrent UCRL is wasteful of the exploratory effort of the agents, because it sends all the agents who have not left the source to the same chain with the most optimistic last edge reward. Seed LSVI and seed TD exhibit diversity and perform identically with seed sampling, which again starts with a very informed prior.

## 4.2 Scaling Up: Cartpole Swing-Up

In this section we extend the algorithms and insights we have developed in the rest of the paper to a complex non-linear control problem. We revisit a variant of the "cartpole" problem of Section 2, but we introduce two additional state variables, the horizontal distance of the cart $x_t$ from the center

$x = 0$ and its velocity, $\dot{x}_t$. The second order differential equation governing the system becomes $\ddot{\phi}_t = \frac{g\sin(\phi_t) - \cos(\phi_t)\tau_t}{\frac{l}{2}\left(\frac{4}{3} - \frac{m}{m+M}\cos(\phi_t)^2\right)}$, $\ddot{x}_t = \tau_t - \frac{m\frac{l}{2}\ddot{\phi}_t\cos(\phi_t)}{m+M}$, $\tau_t = \frac{F_t + \frac{l}{2}\dot{\phi}_t^2\sin(\phi_t)}{m+M}$ [11]. We discretize the evolution of this second order differential equation with timescale $\Delta t = 0.01$. The agent receives a reward of $1$ if the pole is balanced upright, steady in the middle and the cartpole is centered (precisely when $\cos(\phi_t) > 0.95$, $|x_t| < 0.1$, $|\dot{x}_t| < 1$ and $|\dot{\phi}_t| < 1$), otherwise the reward is $0$. We evaluate performance for 30 seconds of interaction, equivalent to 3000 actions. For implementation, we use the DeepMind control suite that imposes a rigid edge at $|x| = 2$ [22].

Due to the curse of dimensionality, tabular approaches to seed sampling quickly become intractable as we introduce more state variables. For a practical approach to seed sampling in this domain we apply the seed TD-ensemble algorithm of Sections 3.2.2 and 3.2.3, together with a neural network representation of the value function. We pass the neural network six features: $\cos(\phi_t), \sin(\phi_t), \frac{\dot{\phi}_t}{10}, \frac{x}{10}, \frac{\dot{x}_t}{10}, \mathbb{1}\{|x_t| < 0.1\}$. Let $f_\theta : \mathcal{S} \to \mathbb{R}^{\mathcal{A}}$ be a $(50, 50)$-MLP with rectified linear units and linear skip connection. We initialize each $Q^e(s, a \mid \theta^e) = \left(f_{\theta^e} + 3f_{\theta_0^e}\right)(s)[a]$ for $\theta^e, \theta_0^e$ sampled from Glorot initialization [2]. After each action, for each agent we sample a minibatch of 16 transitions uniformly from the shared replay buffer and take gradient steps with respect to $\theta^e$ using the ADAM optimizer with learning rate $10^{-3}$ [8]. The parameter $\theta_0^e$ plays a role similar to the prior regularization $\psi$ when used in conjunction with SGD training [9]. We sample noise $z_{e,j} \sim \mathcal{N}(0, 0.01)$ to be used in the shared replay buffer.

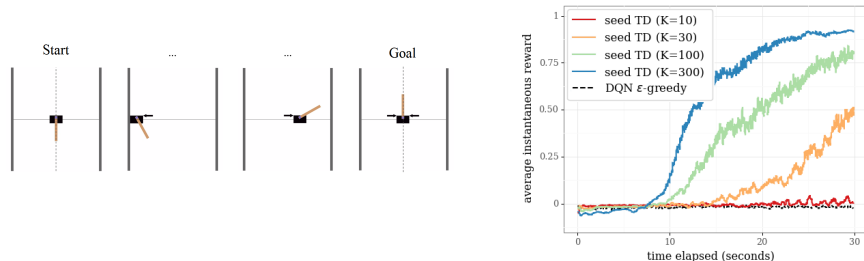

Figure 3: Comparison of seed sampling varying the number $K$ of agents, with a model ensemble size $\min(K, 30)$. As a baseline we use DQN with 100 agents applying $\epsilon$-greedy exploration.

Figure 3 presents the results of our seed sampling experiments on this cartpole problem. Each curve is averaged over 10 random instances. As a baseline, we consider DQN with 100 parallel agents each with 0.1-greedy action selection. With this approach, the agents fail to see any reward over the duration of their experience. By contrast, a seed sampling approach is able to explore efficiently, with agents learning to balance the pole remarkably quickly [1]. The average reward per agent increases as we increase the number $K$ of parallel agents. To reduce compute time, we use seed ensemble with $\min(K, 30)$ models; this seems to not significantly degrade performance.

## 5    Closing Remarks

We have extended the concept of seeding from the non-practical tabular representations to generalized representations and we have proposed an approach for designing scalable concurrent reinforcement learning algorithms that can intelligently coordinate the exploratory effort of agents learning in parallel in potentially enormous state spaces. This approach allows the concurrent agents (1) to adapt to each other's high-dimensional observations via value function learning, (2) to diversify their experience collection via an intrinsic random seed that uniquely initializes each agent's generalized representation and uniquely interprets the common history of observations, (3) to commit to sequences of actions revealing useful information by maintaining each agent's seed constant throughout learning. We envision multiple applications of practical interest, where a number of parallel agents who conform to the proposed framework, can learn and achieve high rewards in short learning periods. Such application areas include web services, the management of a fleet of autonomous vehicles or the management of a farm of networked robots, where each online user, vehicle or robot respectively is controlled by an agent.

## Footnotes

[1]For a demo, see https://youtu.be/kwvhfzbzb0o

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
