[Reviews · NeurIPS 2018]

Reviewer 1



Main ideas of the submission The authors investigate the problem of efficient coordinated concurrent exploration in environments too large to be addressed by tabular, model-based methods. This is a continuation of [1], where the principles of seed sampling were developed for efficient coordinated concurrent exploration, using a tabular model based algorithm. Since the algorithm was only tested on trivial tasks in [1], the authors first demonstrate the effectiveness of this tabular method on a more challenging problem (swinging up and balancing a pole), compared to trivial extensions of known methods (UCB, Posterior sampling) to the concurrent setting. Following that, they suggest a model-free extension to seeding that is based on function approximation with randomized value functions [9] – a concept that facilitates the combination of the seeding principle with generalization. The authors also suggest some concrete algorithms (SLSVI, STD) that support this concept, show that its performance on the trivial examples of [1] is comparable to that of tabular seed sampling, and show its effectiveness on another pole-balancing problem, which is too difficult to be addressed by tabular methods. Strong points • Showing that the tabular seed sampling method suggested in [1] is effective compared to other concurrent exploration methods, even on the more difficult problem of balancing up and stabilizing a pole. • Extending the seeding idea in [1] to complex continuous environments. • Although the authors suggest some concrete algorithm implementations (SLSVI, STD) – the concept itself is very general and can be employed in various versions. • The authors also discuss attempts to lower the computational cost – such as limiting the number of operating models (section 3.2.3). • The suggested method is shown to achieve a performance, which is almost as good as the one achieved by the tabular seeding method from [1], although the current method has a much less informative prior. They also use a high dimensional task to demonstrate that the exploration becomes more efficient as the number of agents increases, and is more efficient than \epsilon-greedy high dimensional exploration. Weak points and suggestions • Although seeding is a promising principle for efficient concurrent exploration, this idea is not backed up by any theoretical analysis and the benefits are only exemplified computationally. Although [14] has developed provable upper exploration cost bounds for the concurrent exploration case, [12] and [9] have only developed such results only for the single-agent case. In a wider perspective, while the current submission and [1] repeatedly claim that they have found three necessary properties for efficient concurrent exploration, there is no supporting theory for that. • The concrete suggested algorithms essentially rely on existing methods (LSVI and TD) for which the computational cost has been well tested in previous work. The only additional cost per agent is the drawing of a random seed, and they also suggest a way to limit the number of active models to save computational costs. This means that essentially – the computational cost of the algorithm itself is the number of active models times the exploration cost for a single agent. I think that this should be noted in the submission, so that the readers are able to compare the computational requirements with previous works. • The seeding method suggested in this paper is quite general, and lacks some guidelines on how to choose the distribution of the seed Z. The authors of [1], make the effort to show that the distribution of the seed Z at the beginning of the episode, is chosen in a specific manner so that at each time - \hat\theta(z,H_t)\sim \theta|H_t (Where \theta|H_t is the posterior estimation of the model parameter \theta, based on the history up to that time H_t, and \hat\theta(z,H_t) is the parameter chosen by an agent based on a seed z and history H_t. Note that the randomness stems from the seed z). However, there is no indication at all on a how to effectively choose the seed Z in the current submission. Although the concept of a randomizing seed is intuitive, a poor choice can ruin the exploration (as the seed acts to modify sampled rewards), which makes this work incomplete. • The agents all explore the same MDP, but a more interesting situation is concurrent exploration on a set of different environments with some shared properties (as investigated in [14]). Since the environments are not identical, there are some properties that agents should share and some that they should not, and that is another difficulty that the agents must overcome. This also closely relates to the problem of transfer learning. • The two example tasks discussed in the submission (balancing up a pole) are not complicated enough in the sense that the environment is deterministic. It would be interesting to see how the algorithms operate in such a system where, for example, a noisy wind is introduced. Furthermore, the task of steadying a pole can surely benefit from concurrent exploration: When effective, the concurrent exploration can make the learning converge faster. However, there are surely more interesting tasks where concurrent exploration and cooperation between agents is necessary for the achievement of the overall target, and it would be interesting to see if the seeding-method operates well in these tasks too. Technical details: • In the example task of balancing up a pole – the differential equations for the cart and mass are taken from [18] in their references, which seems to direct to the following paper: Cohen, Michael A., and Stephen Grossberg. "Absolute stability of global pattern formation and parallel memory storage by competitive neural networks." IEEE transactions on systems, man, and cybernetics 5 (1983): 815-826. Even when taking the friction \mu to be zero, the equations does not seems to match the ones presented in the submission. • Line 79: There is no detailed explanation regarding the state-actions that receive a reward of 1. For example, it is unclear whether the pole must be balanced with a zero velocity, or must just be in an upright position. In fact, the authors explain it more clearly in the second pole example (Note 1 on page 7 of the submission), but it should be explained here as well. • Line 138: The paraphrasing should be changed, as it is unclear whether the samples perturbed by an agent are also visible to other agents. • The formula for computing \tilde\theta_h used in the SLSVI of section 3.2.1 is said to run for h=H-1,...,0. Since we consider a finite horizon problem in which each agent only takes H actions at each episode – wouldn’t it make more sense for this equation, before the m-th action, to run for only h=H-1,...,m? Since LSVI aims to find the parameters for which the Q-function is the closest to satisfying the finite horizon Bellman equations, restricting this equation to h=H-1,...,m means that at the m-th action, this would be the next choice considering that there are only H-m transitions left. • The minimization operation present in the formula for computing L(\theta) following eq. 209 seems to be a typo, since a minimization operation is the purpose of the SGD. Furthermore, there seems to be a discount factor added to this formula – which means that this SGD is used to solve the discounted MDP problem, and not the finite horizon one as stated before. However, this is never specified within the text. • The example tasks of section 4.1 employ the generalized seeding algorithms, but it is not specified how the random seed is set (meaning, the distribution of the random noise terms Z that are added to the rewards is not mentioned) – and thus, these results cannot be reproduced by readers. • Example 4.2 – it is unclear what the "instantaneous reward" axis in figure 3 is. Is it the reward received per agent divided by one second, or is it the reward per time ste

Reviewer 2



This paper proposes an extension of seed sampling for coordinated exploration to enable scalable learning to high dimensional problems. The first set of experimental results in Section 2, feels like an incremental extension of a previous publication (cited by the authors as [1]) as it is not justified well as a direct contribution to the topic of this specific paper and uses the algorithm proposed in the previous paper not the new algorithms proposed in this paper. This section also assumes a significant amount of prior knowledge of the previous work. I would recommend this space would have been better utilised to give a detailed description of seed sampling so that the paper is a self contained reference on scalable seed sampling. The experiment in Section 2, uses a reasonable set of baselines with good alignment with the conditions the authors argue are necessary for concurrent exploration. However, the UCRL method used is a weak baseline and a fairer comparison would be to parallel UCB. As a starting point for exploring the literature on these methods, I would suggest the papers: - T. Cazenave and N. Jouandeau, “On the Parallelization of UCT,” 2007 - G. M. J.-B. Chaslot, M. H. M. Winands, and H. J. van den Herik, “Parallel Monte-Carlo Tree Search.” 2008 - A. Bourki, G. M. J.-B. Chaslot, M. Coulm, V. Danjean, H. Doghmen, J.B. Hoock, T. H´erault, A. Rimmel, F. Teytaud, O. Teytaud, P. Vayssi`ere, and Z. Yu, “Scalability and Parallelization of Monte-Carlo Tree Search” 2010 In Section 3, the authors propose three algorithms that are later tested (seed LSVI, seed TD and seed ensembles). They then argue that this method could also be applied to policy function approximation, which seems feasible but is not empirically demonstrated. I would argue that, given the motivation of scalable approaches, an evaluation of using seed sampling with a modern policy gradient algorithm would have been more insightful than seed LSVI as LSVI is known not to scale well. The results in Section 4.1 are promising and a useful comparison for understanding the effect of the changes proposed in this paper. The authors have again done a great job of aligning the example problem domains and baseline comparisons with the beneficial features of the method they are proposing. However, Section 4.2 is less impressive and (as the single large scale experiment in a paper on scaling up an existing method) significantly weakens the contribution of the paper. Specifically, I would note that this section does not provide sufficient details of the hyperparameters used to reproduce this experiment and compares against a weak baseline. Why hold the number of agents for DQN fixed whilst modifying the number for the proposed method? Why only compare against a baseline that never learns to improve above a random policy? Are there no existing methods (perhaps ones that are significantly more sample inefficient) that can learn to act in this problem domain? During the rebuttal period, it would be useful for the authors to discuss the motivation for including the experiments of Section 2 within this paper and to justify the choice of baseline or proposed changes for the results in Section 4.2. -- Having carefully considered the authors' response, I have raised my view of the paper to just above the acceptance threshold. Particularly based upon the promise in the response to fully document all hyperparameters in the appendix and to release the source code. These efforts will greatly improve the reproducibility of the paper, and help with the uptake of the work by others to further build upon the results presented.

Reviewer 3



The paper proposes an extension of previous work on seeding to large/continuous problems for concurrent RL agents. This paper's idea builds on previous work on seed sampling and randomized value function learning. In general, this is an interesting problem to tackle given a recent exciting interest in this topic. The extension of the previous seed sampling approach for concurrent RL to higher-dimensional problems via the use of function approximation is neccessary and important. In overall, the paper is well written and easy to follow. The discussion on motivations, background and related work are sufficient. The descriptive example and sanity check example are good to understand the idea. The experiment in 4.2 is though on toy domain, but interesting to see how it works in reality. 1. Section 2: it would be more complete if the authors also present background on the work: randomized value function learning. This piece of work is also one of the building block of the proposed idea in this paper. 2. In Section 3.1: It would be clearer if the authors elaborate how the noise term z for diversification can be used to purturb the same observation in the buffer. This might be related to the comment 1 too, as the foundation on randomized value function learning is not introduced. 3. Section 3.2.1: Seed LSVI is related to randomized LSVI in [12]? How it is linked to the previous seeding method in [1]? More discussion and analysis on sampling of z and its use in value function updates are needed. Similar to the extension of Seed TD and Seed Ensemble. More analysis on their committment, Diversification, and Adaptation: why, and how. This might be related to the comment 1 too, as the foundation on randomized value function learning is not introduced. Though refering the the paper [9] but there is still gap in understanding the proposed idea that is based on the idea of randomized value function learning in [9]. 4. Section 3.2.3: The description of the Seed Ensemle is so short that it's not clear to see how ensemble model would be used in each step for each agent. 5. The idea has been proved in a toy domain, but it would be much nicer and make the paper stronger if more experiments on larger domains or more complex domains as discussed in Section 5 for future work can be added. Given the above concerns, I thought that the paper has interesting ideas but seems to be premature to publish at NIPS. ------------------------------------------- I have read the response and decide to raise my rating to borderline (5)